



Soil and plant $\delta^{15}$N have a different response to experimental
warming: A global meta-analysis
Kaihua Liao[1,2*], Xiaoming Lai[1,2], Qing Zhu[1,2,3*]
[1]Key Laboratory of Watershed Geographic Sciences, Nanjing Institute of Geography
and Limnology, Chinese Academy of Sciences, Nanjing 210008, China
[2]University of Chinese Academy of Sciences, Beijing 100049, China
[3]Jiangsu Collaborative Innovation Center of Regional Modern Agriculture &
Environmental Protection, Huaiyin Normal University, Huaian 223001, China

[*] Corresponding author. Tel.: +86 25 86882139; fax: +86 25 57714759.
*E-mail addresses:* khliao@niglas.ac.cn (Kaihua Liao); qzhu@niglas.ac.cn (Qing Zhu)



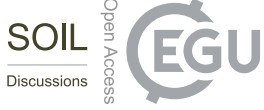

**Abstract.** The [15]N natural abundance composition ($\delta^{15}$N) in soils or plants is a useful
tool to indicate the openness of ecosystem N cycling. This study was aimed to
evaluate the influence of the global warming on soil and plant $\delta^{15}$N. We applied a
global meta-analysis method to synthesize 79 and 76 paired observations for soil and
plant $\delta^{15}$N from 20 published studies, respectively. Results showed that the mean
effect sizes of the soil and plant $\delta^{15}$N under experimental warming were -0.524 (95%
CI: -0.987 to -0.162) and 0.189 (95% CI: -0.210 to 0.569), respectively. This indicated
that soil and plant $\delta^{15}$N had negative and positive responses to warming at the global
scale, respectively. Experimental warming significantly ($p < 0.05$) decreased soil $\delta^{15}$N
in Alkali soil, grassland/meadow, and under air warming, whereas it significantly ($p <$
0.05) increased soil $\delta^{15}$N in neutral soil. Plant $\delta^{15}$N significantly ($p < 0.05$) increased
with increasing temperature in neutral soil and significantly ($p < 0.05$) decreased in
alkali soil. Latitude did not affect the warming effects on both soil and plant $\delta^{15}$N.
However, the warming effect on soil $\delta^{15}$N was positively controlled by the mean
annual temperature, which is related to the fact that the higher temperature can
strengthen the activity of soil microbes. The effect of warming on plant $\delta^{15}$N had
weaker relationships with environmental variables compared with that on soil $\delta^{15}$N.
This implied that soil $\delta^{15}$N tended to be more efficient in indicating the openness of
global ecosystem N cycling than plant $\delta^{15}$N.



## 1 Introduction


Nitrogen (N) is one of the most important nutrient elements for plant growth and the
key limiting factors for vegetation productivity (McLay et al., 2001; Zhu et al., 2018;
Lu et al., 2020). On the one hand, if the available N in the soil is insufficient, it will
damage and weaken the ecosystem service function, including the supply of primary
material products, water conservation, climate regulation, etc. (Averill and Waring,
2018). On the other hand, if the available N in the soil is over supplied, it will also
damage the structure and function of the ecosystem, resulting in a series of
environmental problems such as soil acidification and imbalance of ecosystem
nutrient (Schrijver et al., 2008). The intermediate products of the N cycle process,
such as nitrate nitrogen ($NO_3^- - N$), nitrous oxide ($N_2O$) and nitric oxide (NO), may
also cause eco-environmental problems such as eutrophication of water body and
aggravation of greenhouse effect (Liao et al., 2019). Therefore, it is of great
significance to reveal the openness of the ecosystem N cycle process for
understanding the plant N fixation and long-term trend of N cycling and protecting
the eco-environment (Wang et al., 2014; Wu et al., 2019).
The $^{15}N$ natural abundance composition ($\delta^{15}N$) in soils or plants (leaves, shoots,
fine roots and litter) becomes a useful tool to indicate the openness of ecosystem N
cycling (Robinson, 2001). This is because the lighter $^{14}N$ always preferentially loses
from the ecosystem. Thus, the heavier $^{15}N$ gradually enriches in the ecosystem,
resulting in the isotopic fractionation effect (Aranibar et al., 2004). The larger the
$\delta^{15}N$ value, the higher degree of openness of N cycling. A large number of studies





have confirmed that climate was the main factor regulating the soil and plant $\delta^{15}$N
(Craine et al., 2015; Soper et al., 2015). Previous studies have demonstrated that
precipitation had a negative effect on soil and plant $\delta^{15}$N from in-situ evidences to
cross-sites syntheses (Swap et al., 2004; Soper et al., 2015). However, the influence of
temperature on soil and plant $\delta^{15}$N remained controversial. Some studies have showed
that soil and plant $\delta^{15}$N increased with temperature (Amundson et al., 2003; Craine et
al., 2015), while others have indicated that $\delta^{15}$N decreased with temperature (Cheng et
al., 2009; Sheng et al., 2014) or even there was no correlation between them (Yang et
al., 2013). The various studies suggested that the responses of soil and plant $\delta^{15}$N to
warming were very complex and not well understood. In addition to climate factor,
soil and plant $\delta^{15}$N are affected by a variety of other environmental factors, such as
vegetation type, topography, soil properties and management practices (Gurmesa et al.,
2017; Wang et al., 2019). However, we know little about the influences of
environmental factors on the warming effect on ecosystem N cycling, in terms of soil
and plant $\delta^{15}$N.
Soil warming experiment was often conducted to study the effect of warming on
the ecosystem N cycling at site scale (Schindlbacher et al., 2009). At present, the
effect of experimental warming on soil and plant $\delta^{15}$N has not been studied on a
global scale. The objectives of this study were to: (i) detect the effect of experimental
warming on the soil and plant $\delta^{15}$N based on a global meta-analysis of 20 studies; and
(ii) identify the main factors influencing the warming effect on the soil and plant $\delta^{15}$N.
Specifically, we hypothesized that soil and plant $\delta^{15}$N have a different response to



experimental warming.

## 2 Materials and methods

### 2.1 Source of data and selection criteria

Peer-reviewed journal articles and dissertations related to soil and plant $\delta^{15}N$ under
experimental warming were searched using Web of Science and China National
Knowledge Infrastructure (CNKI, http://www.cnki.net) until March 31, 2020 (Tab. 1).
The keywords used for the literature search were related to: "nitrogen isotope
composition", "experimental warming" and "ecosystems nitrogen cycling".
Our criteria were as follows: at least one of the target variables must be contained,
including soils (different fractions, e.g., sand, silt, clay, aggregate and bulk soil) and
plants (leaves, shoots, roots and litters) $\delta^{15}N$; studies with temperature gradients were
excluded and only field warming experimental studies were included; only data from
control and warming treatments were applied for multifactor experiments; means,
standard deviations (SD) (or standard errors (SE)) and sample sizes were directly
provided or could be calculated from the studies; if one article contained soil or plant
$\delta^{15}N$ in multiple years, only the latest results were applied since the observations
should be independent in the meta-analysis (Hedges et al., 1999).

### 2.2 Data extraction and statistical analysis

In total, 20 published papers were selected from more than 50 published papers. The
locations of warming experiments were presented and their site information is listed
in Tab. 1. For each study, the means, the statistical variation (SE or SD) and the
sample size values for treatment and control groups were extracted for each response





variable ($\delta^{15}$N). In addition to $\delta^{15}$N, the latitude, longitude, altitude, soil pH,
vegetation type, mean annual precipitation (MAP) and mean annual temperature
(MAT) were also extracted if they can be obtained (Tab. 1). All data were extracted
from tables or digitized from graphs with the software GetData v2.2.4
(http://www.getdata-graph-digitizer.com). A total of 79 and 76 paired observations for
soil and plant $\delta^{15}$N were obtained, respectively.

The METAWIN 2.1 software (Sinauer Associates Inc., Sunderland, MA, USA)

(Rosenberg et al., 2000) was used to perform meta-analysis in this study. The Hedges'
*d* value was used as the effect size (Hedges et al., 1999). The absolute *d* value
indicated the magnitude of the treatment impact. Positive or negative *d* values
represented an increase or decrease effect of the treatment, respectively. Zero meant
no difference between treatment and control groups. The mean effect size and 95%
bootstrap confidence intervals (CI) were then generated. If the 95% CI values of *d* did
not overlap zero, the effects of experimental warming on $\delta^{15}$N were considered
significant at $p < 0.05$. We used a random effects model to test whether warming had a
significant effect on $\delta^{15}$N. To examine whether experimental conditions alter the
response direction and magnitude of soil and plant $\delta^{15}$N, observations were further
divided into subgroups according to the soil acidity-alkalinity (acid (pH < 6.5),
neutral (6.5 < pH < 7.5), and alkali (pH > 7.5)), vegetation types (forest/shrub,
moss/lichen, and grassland/meadow), and warming treatments (soil warming, air
warming, and both soil and air warming). A random effects model with a grouping
variable was used to compare responses among different subgroups. Linear regression





analyses were applied to assess the relationships between the Hedges' $d$ values and
environmental factors (i.e., latitude, altitude, MAT and MAP).
**3 Results**
Across all sites, the mean effect sizes of the soil and plant $\delta^{15}N$ under experimental
warming were -0.524 (95% CI: -0.987 to -0.162) and 0.189 (95% CI: -0.210 to 0.569),
respectively (Fig. 1). Experimental warming significantly ($p < 0.05$) decreased soil
$\delta^{15}N$ in Alkali soil (mean effect size = -2.484; 95% CI: -2.931 to -2.060),
grassland/meadow (mean effect size = -0.609; 95% CI: -1.076 to -0.190), and under
air warming (mean effect size = -0.652; 95% CI: -1.081 to -0.273), whereas it
significantly ($p < 0.05$) increased soil $\delta^{15}N$ in neutral soil (mean effect size = 0.359;
95% CI: 0.078 to 0.620) (Fig. 2). However, experimental warming did not
significantly ($p > 0.05$) change soil $\delta^{15}N$ in acid soil (mean effect size = -1.084; 95%
CI: -3.588 to 9.211), forest/shrub (mean effect size = -0.179; 95% CI: -1.619 to 1.641),
and under soil warming (mean effect size = 0.189; 95% CI: -1.304 to 2.041). In
addition, experimental warming significantly ($p < 0.05$) increased plant $\delta^{15}N$ in
neutral soil (mean effect size = 3.157; 95% CI: 1.529 to 6.967), whereas it
significantly ($p < 0.05$) decreased plant $\delta^{15}N$ in alkali soil (mean effect size = -1.930;
95% CI: -2.325 to -1.573). However, experimental warming did not significantly ($p >$
0.05) change plant $\delta^{15}N$ under other experimental conditions.

For soil and plant $\delta^{15}N$, their responses to experimental warming did not correlate

well with latitude ($p = 0.268$ and $p = 0.160$, respectively) (Fig. 3ab). However, the
Hedges' $d$ values of soil $\delta^{15}N$ decreased significantly with altitude ($p < 0.001$) (Fig. 3c)



and increased significantly with MAT ($p < 0.001$) and MAP ($p < 0.001$) (Fig. 3eg). In
addition, the Hedges' $d$ values of plant $\delta^{15}N$ were also found to increase significantly
with MAP ($p < 0.001$) (Fig. 3h). However, the responses of plant $\delta^{15}N$ to experimental
warming did not correlate well with altitude ($p = 0.109$) and MAT ($p = 0.002$) (Fig.
3df).
**4 Discussion**
Soil and plant $\delta^{15}N$ showed a different pattern under experimental warming at the
global scale, with a significant decreasing trend in soil $\delta^{15}N$ and an increasing trend in
plant $\delta^{15}N$. This is somewhat inconsistent with previous findings. Chang et al. (2017)
observed that soil and plant $\delta^{15}N$ values decreased under warming in the Tibetan
permafrost. However, Zhang et al. (2019) found that the warming treatment
significantly increased soil and plant $\delta^{15}N$ in a subtropical forest. The various studies
suggest that soil and plant $\delta^{15}N$ are controlled by interactive effects of N fixation and
mineralization. At the global scale, $\delta^{15}N$ of N input (~ 0) is generally lower than that
of soil, so greater N fixation or higher N input (deposition and fertilization) under
warming can result in a lower soil $\delta^{15}N$ (Rousk and Michelsen, 2017). Plants mainly
absorb the inorganic nitrogen in the soil, while increasing temperature can cause
higher N mineralization rates and a subsequent increase in plant $\delta^{15}N$ (Swap et al.,

2004).

Soil pH has an important influence on nitrification, denitrification and $N_2O$
emissions from soils (Kyveryga et al., 2004). The results in this study showed that
when the soil was acidic or alkaline, the mean effect sizes of soil and plant $\delta^{15}N$ under



163 warming were negative, while when the soil was neutral, they were positive (Fig. 2ab).

164 Compared with acid or alkaline conditions, the near neutral conditions are more

165 suitable for the biological activities of heterotrophic denitrifying bacteria (Simek and

166 Cooper, 2002). Therefore, the denitrification activity is usually higher under neutral

167 conditions, resulting in an enrichment of soil and plant N pools with $^{15}$N. Vegetation

168 type has a limited effect on soil and plant $\delta^{15}$N under warming, showing that the mean

169 effect sizes of $\delta^{15}$N in forest/shrub were slightly higher than those in moss/lichen and

170 grassland/meadow (Fig. 2cd). This may be related to the differences in altitude, MAP

171 and MAT among the three vegetation types (Tab. 1). Warming treatment was found to

172 have a substantial effect on soil and plant $\delta^{15}$N. The mean effect size of soil $\delta^{15}$N

173 under soil warming was higher than that under air warming, while the mean effect

174 sizes of plant $\delta^{15}$N were sequenced as soil warming > air warming > both soil and air

175 warming (Fig. 2ef). Salmon et al. (2016) have found that soil warming can increase N

176 availability by stimulating mineralization of organic matter in the warmed actively

177 layer. However, air warming would only impact aboveground temperatures and has a

178 weak effect on soil and plant N pools which are related to $\delta^{15}$N (Pardo et al., 2006). In

179 addition, when the two warming treatments were applied together, the inorganic N

180 availability and $\delta^{15}$N signature of plants decreased due to the negative interaction

181 between air warming and soil warming treatments (Salmon et al., 2016).

182   The warming effects on soil and plant $\delta^{15}$N had weak correlations with latitude

183 (Fig. 3), although soil and plant $\delta^{15}$N were negatively correlated with latitude at the

184 global scale ($p \leq 0.001$) as found in Mayor et al. (2015). However, the warming effect



on soil $\delta^{15}$N was significantly influenced by altitude, MAT and MAP. Among these,
the strongest correlation was observed for MAT. Temperature has been demonstrated
to be a key factor to regulate the soil $\delta^{15}$N by influencing the processes of N
mineralization, nitrification and denitrification. The higher temperature can strengthen
the activity of soil microbes and thereafter increase the N uptake for plants and soil N
loss from ammonia volatilization and gas N emissions, and thereby more $^{15}$N-enriched
retains in soils (Wang et al., 2019). High $d$ values of soil $\delta^{15}$N corresponded to MAT
of about 20 $^{\circ}$C, which was the most suitable temperature for nitrification and
denitrification. However, warming had a substantial negative impact on soil $\delta^{15}$N
when MAT decreased to around -5 $^{\circ}$C. Recently, Rousk et al. (2018) also found that
the increase of temperature in the Arctic promoted the biological N fixation, which
can decrease the soil $\delta^{15}$N. The decrease of $d$ values of soil $\delta^{15}$N with increasing
altitude and decreasing MAP in this study might be caused by the positive response of
$d$ values to MAT. The relationships between the $d$ values and environmental variables
for plant $\delta^{15}$N were weaker than those for soil $\delta^{15}$N (Fig. 3). The possible reason is
that several other factors (e.g., plant N concentrations and species richness) might
co-regulate plant $\delta^{15}$N (Wu et al., 2019). This implied that soil $\delta^{15}$N was more
efficient in indicating the openness of ecosystem N cycling than plant $\delta^{15}$N at the
global scale.

Although the present study provided a global meta-analysis of the responses of

$\delta^{15}$N to experimental warming, the magnitude of these responses might be uncertain.
For example, a small number of observations were obtained in moss/lichen under soil



warming and both soil and air warming treatments, which would affect the results of
meta-analysis. In addition, the duration of field exposure to warming and increase in
temperature may also have an impact on ecosystem N cycling (Dawes et al., 2017).
Future research should take various experimental durations and temperature increases
into account when investigating the warming effects on $\delta^{15}N$.
**6 Conclusions**
Our global meta-analysis indicated a significant decreasing trend in soil $\delta^{15}N$ and an
increasing trend in plant $\delta^{15}N$ under experimental warming. Latitude did not affect the
warming effects on $\delta^{15}N$. However, the warming effect on soil and plant $\delta^{15}N$ was
related to soil acidity-alkalinity, vegetation type, warming treatment, altitude, MAT
and MAP. The effect of warming on soil $\delta^{15}N$ was better correlated with
environmental variables compared with that on plant $\delta^{15}N$. Our findings should be
useful for understanding the underlying mechanisms of the response of ecosystem N
cycling to global warming.
**Data availability.** The data that support the findings of this study are available from
the corresponding author upon request.
**Author contributions.** KL and QZ designed this study, KL and XL performed the
meta-analysis, KL and QZ obtained funding, and KL and XL wrote the paper with
contributions from QZ.
**Competing interests.** The authors declare that they have no conflict of interest.
**Acknowledgements.** We thank two anonymous reviewers and editor for their efforts
on this paper. Support for this research was provided by the National Natural Science



Foundation of China and by Chinese Academy of Sciences.
**Financial support.** This study was financially supported by the National Natural
Science Foundation of China (41771107), the Key Research Program of Frontier
Sciences, Chinese Academy of Sciences (QYZDB-SSW-DQC038), and the Youth
Innovation Promotion Association, Chinese Academy of Sciences (2020317).
**Review statement.** This paper was reviewed by two anonymous referees.

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



**Table 1:** Site characteristics from a global meta-analysis of 20 studies.

| References | Country/Region | Vegetation types | Latitude | Longitude | Altitude (m a.s.l) | MAT (°C) | MAP (mm) |
|---|---|---|---|---|---|---|---|
| Anadon-Rosell et al. (2017) | Spain | Subalpine shrub | 41.39 °N | 2.17 °E | 2250 | 3 | 1146.4 |
| Zhang et al. (2019) | China | *C. lanceolata* seedlings | 26.32 °N | 117.6 °E | 300 | 19.1 | 1670 |
| Lim et al. (2019) | Sweden | Boreal forests | 64.12 °N | 19.45 °E | 310 | 2.4 | 600 |
| Deane-Coe et al. (2015) | USA | Tundra mosses | 63.88 °N | 149.23 °W | 700 | -2.7~-1 | 138~228 |
| Bijoor et al. (2008) | USA | Turfgrass lawn | 33.7 °N | 117.7 °W | 30 | 18.6 | 352 |
| Chang et al. (2017) | China | Alpine meadow | 34.73 °N | 92.89 °E | 4750 | -5.3 | 269.7 |
| Gonzalez-Meler et al. (2017) | Brazil | Grasslands | 21.17 °S | 47.86 °W | 578 | 21.5 | 1100 |
| Natali et al. (2012) | USA | Shrubs, sedges and mosses | 63.88 °N | 149.23 °W | 700 | -1 | 178~250 |
| Munir et al. (2017) | Canada | Shrubs, mosses and trees | 55.27 °N | 112.47 °W | | | |
| Salmon et al. (2016) | USA | *Eriophorum vaginatum* | 63.88 °N | 149.23 °W | 700 | -1.45 | 200 |
| Rui et al. (2011) | China | Alpine meadow | 37.62 °N | 101.2 °E | 3200 | -2 | 500 |
| Aerts et al. (2009) | Sweden | Shrubs, mosses and trees | 68.35 °N | 18.82 °E | 340 | 0.5 | 303 |
| Cheng et al. (2011) | USA | Tallgrass prairie | 34.98 °N | 97.52 °W | | 16 | 911.4 |
| Dawes et al. (2017) | Switzerland | Alpine treeline | 46.77 °N | 9.87 °E | 2180 | 9.2 | 444 |
| Schaeffer et al. (2013) | Greenland | Prostrate dwarf-shrub herb tundra | 76 °N | 68 °W | | 4~8 | <200 |
| Schnecker et al. (2016) | Austria | Spruce forest | 47.58 °N | 11.64 °E | 910 | 6.9 | 1506 |
| Hudson et al. (2011) | Canada | Heath, willow and meadow | 78.88 °N | 75.78 °W | | 8.6~10.4 | |
| Lv et al. (2018) | China | *Cunninghamia lanceolata* juveniles | 26.32 °N | 118 °E | | 19.1 | 1585 |
| Zhao et al. (2016) | China | Alpine meadow | 37.48 °N | 101.2 °E | 3200~3250 | -1.7 | 600 |

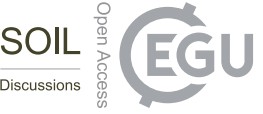

| Peng (2017) | China | Alpine meadow | 34.73 °N | 92.89 °E | 3200~4800 | -5.03 | 267.4~426.3 |

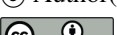



## List of Figures:

**Figure 1:** Effect sizes of the experimental warming on soil and plant $\delta^{15}N$ from a global meta-analysis of 20 studies. The error bars indicate effect sizes and 95% bootstrap confidence intervals (CI). The warming effect was statistically significant if the 95% CI did not bracket zero. The sample size for each variable is shown next to the bar.

**Figure 2:** Factors influencing the effect sizes of the soil and plant $\delta^{15}N$ under experimental warming from a global meta-analysis of 20 studies. The error bars indicate effect sizes and 95% bootstrap confidence intervals (CI). The warming effect was statistically significant if the 95% CI did not bracket zero. The sample size for each variable is shown next to the bar.

**Figure 3:** Relationships between the Hedges' *d* values of soil and plant $\delta^{15}N$ with the latitude, altitude, mean annual temperature (MAT) and mean annual precipitation (MAP) under experimental warming.





Figure 1

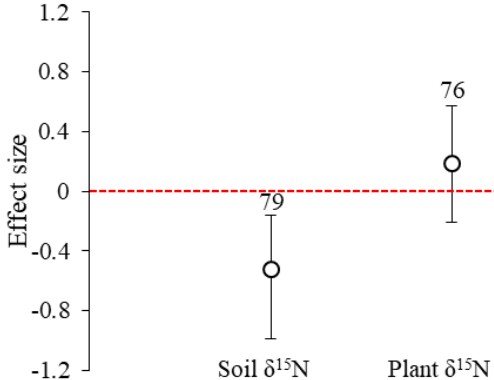



Figure 2

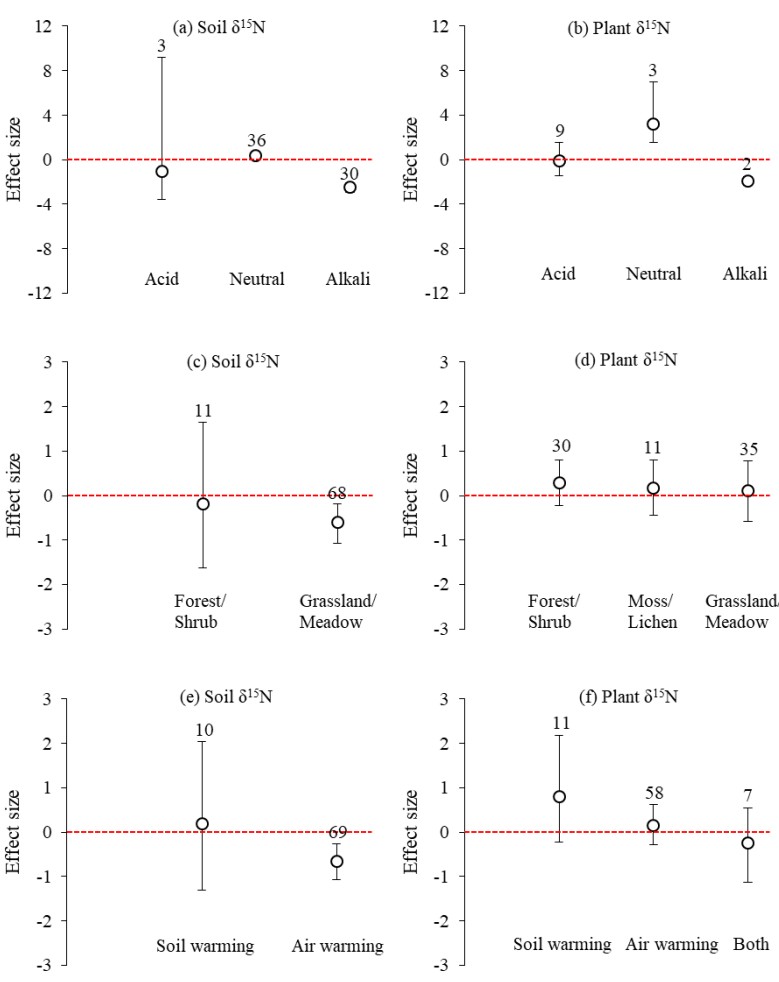



Figure 3

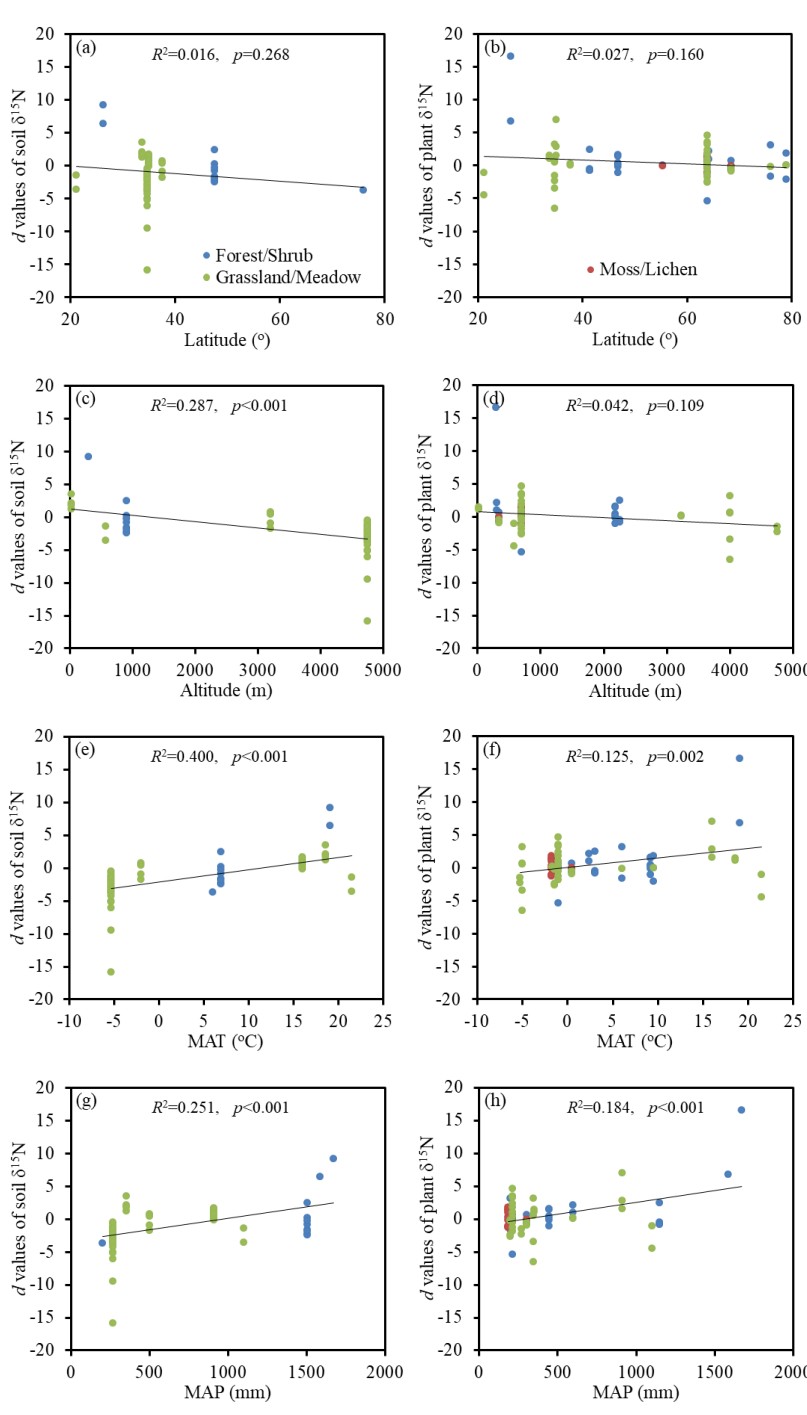