# Peer review of "Soil and plant $\delta^{15}$N have a different response to experimental warming: A global meta-analysis"

_SOIL, 2021_

## Author Comment (AC1)

**Response to CC1**

If you take a look at the syntheses we have done, soil δ15N also appears to reflect the degree of decomposition of the organic matter. δ15N increases with processing. Warmer sites have soil N that is elevated in 15N, but has lower C:N. Once you control for C:N, there is little pattern in 15N across temperature gradients. similar interpretations could be applied here. You need to take a look at the syntheses and reviews we have done on how to interpret plant and soil 15N. there are important data here, but interpretation is important too.

Answer: Thank you very much for your efforts on our paper submitted to the "Soil" (Manuscript ID soil-2021-40). We have checked the manuscript and revised it according to the comments carefully.

In the Introduction section of the revised paper, we have stated that the larger the $\delta^{15}N$ value, the higher degree of openness of N cycling. In addition, soil $\delta^{15}N$ also appears to reflect the degree of decomposition of the organic matter, showing that $\delta^{15}N$ increases with processing (Craine et al., 2015) (P4L58-60).

In the Discussion section, we have indicated that warmer sites have soil N that is elevated in $^{15}N$, but has lower C:N. Once C:N is controlled, there is little pattern in $^{15}N$ across temperature gradients. In other words, the relationship between soil $\delta^{15}N$ and climate is indirect, and mediated through climate effects on soil properties (e.g., the concentrations of organic carbon and clay) (Craine et al., 2015) (P10L207-211).

Finally, the relationships between the *d* values and environmental variables for plant $\delta^{15}N$ were weaker than those for soil $\delta^{15}N$ (Fig. 3). The possible reason is that several other factors (e.g., plant N concentrations and species richness) might co-regulate plant $\delta^{15}N$ (Wu et al., 2019). This is consistent with the study of Craine et al. (2009), who found different inflection points in soil and plant $\delta^{15}N$ relationships with MAT. In addition, plants are generally depleted in $^{15}N$ relative to soils (P11L222-224).

---

## Author Comment (AC2)

**Response to RC1**

1. In this paper, few literatures (only 20 papers) were selected for meta-analysis. Fortunately, there are 79 and 76 paired observations for soil and plant δ15N, and I believe that these samples are enough for meta-analysis. However, the manuscript lacks of important information on soil, such as organic matter content.

Answer: Thank you very much for your efforts on our paper submitted to the "Soil" (Manuscript ID soil-2021-40). We have checked the manuscript and revised it according to the comments carefully.

In the revised paper, we have provided the soil information, including organic matter content, soil type, and soil pH (Table 1). In addition, we also re-conducted the meta-analysis to investigate the influences of soil texture, warming period and increase in temperature on the warming effects on soil and plant $\delta^{15}N$ (Fig. 2). However, we did not consider the effect of soil organic matter content since only four literatures provided the organic matter content values.

2. The Introduction section still needs to be improved to clarify the key scientific issues. Such as, one of purpose of this study is to identify the main factors influencing the warming effect on the soil and plant δ15N. In fact, since a relatively large number of investigators from around the world were involved in the 20 studies that were selected, there is no indication that all these studies used similar protocols, and coordinated their experimental conditions. For example, we know that soil water content has an impact on soil and plant δ15N, and also impacts the analysis results. The authors should check what is the cause of the differences between soil and plant δ15N?

Answer: In the revised manuscript, we have clarified that the objectives of this study were to: (i) detect the effect of experimental warming on the soil and plant $\delta^{15}N$ based on a global meta-analysis of 20 studies; and (ii) identify the main factors influencing the warming effect on the soil and plant $\delta^{15}N$. Specifically, we hypothesized that soil $\delta^{15}N$ is a better indicator of ecosystem N cycling than plant

$\delta^{15}$N (P5L81-82).

Our criteria were as follows: at least one of the target variables was contained, including soils (different fractions, e.g., sand, silt, clay, aggregate and bulk soil) and plants (leaves, shoots, roots and litters) $\delta^{15}$N; studies with climate gradients (space-time substitution) were excluded and only field warming experimental studies were included; only data from control and warming treatments were applied for multifactor experiments; means, standard deviations (SD) (or standard errors (SE)) and sample sizes were directly provided or could be calculated from the studies; if one article contained soil or plant $\delta^{15}$N in multiple years, only the latest results were applied since the observations should be independent in the meta-analysis (Hedges et al., 1999) (P5L90-99).

I agree with your comment that soil water content has an impact on soil and plant $\delta^{15}$N, and also impacts the analysis results. However, in the revised manuscript, we did not consider the effect of soil water content on the warming effects on soil and plant $\delta^{15}$N. This is due to the fact that soil water content has strong temporal and spatial variability, so it is difficult to control soil moisture in warming experiment. Nevertheless, in the revised paper, we re-conducted the meta-analysis to investigate the influences of soil texture, warming period and increase in temperature on the warming effects on soil and plant $\delta^{15}$N (Fig. 2).

3. The M & M section is a little rough and please reorganize this section to make it more clearly. Please provide detailed soil information which appeared in this section.

Answer: In the revised paper, we have provided detailed soil information, including organic matter content, soil type, and soil pH (Table 1). In addition, we also reorganized this section to make it more clearly.

4. The Discussion section, which is the most important part, is relatively weak because the contents and explanations are not well organized and are still not convincing. Less descriptive of results and more mechanism discussion are

encouraged in this section. More references should be added to the discussion section.

Answer: In the revised manuscript, we have added more references in the discussion section. In addition, we also re-conducted the meta-analysis to investigate the influences of soil texture, warming period and increase in temperature on the warming effects on soil and plant $\delta^{15}$N. From Fig. 2gh, the finer the soil texture, the more significant the positive effect of warming on soil and plant $\delta^{15}$N. The possible reason is that the finer the soil texture, the stronger the adsorption of various ions on the soil and the smaller the leaching loss of the soil, resulting in the greater the residual amount of $^{15}$N in the soil (Webster et al., 1986). In addition, the longer warming period and the greater increase in temperature resulted in the more negative effect of warming on soil $\delta^{15}$N (Fig. 2ik). Chang et al. (2017) deduced that N fixation was greater under warming and consequently resulted in a lower soil $\delta^{15}$N (P10L189-196).

Figure 2:

[Figure]

[Figure]

Temperature has been demonstrated to be a key factor to regulate the soil $\delta^{15}N$ by influencing the processes of N mineralization, nitrification and denitrification (Craine et al., 2015). The higher temperature can strengthen the activity of soil microbes and thereafter increase the N uptake for plants and soil N loss from ammonia volatilization and gas N emissions, and thereby more $^{15}N$-enriched retains in soils (Wang et al., 2019). Craine et al. (2015) also proposed that warmer sites have soil N that is elevated in $^{15}N$, but has lower C:N. Once C:N is controlled, there is little pattern in $^{15}N$ across temperature gradients. In other words, the relationship between soil $\delta^{15}N$ and climate is indirect, and mediated through climate effects on soil properties (e.g., the concentrations of organic carbon and clay) (P10L204-211).

5. Line 12: The nitrogen-15 (15N)...

Answer: In the revised paper, "The $^{15}N$" has been replaced by "The nitrogen-15 ($^{15}N$)" (P2L12).

6. Line 14: global warming -> experimental warming

Answer: Sorry for this confusion. In the revised paper, "global warming" has been replaced by "experimental warming" (P2L14).

7. Line 15: for -> of

Answer: In the revised paper, "for" has been replaced by "of" (P2L16).

8. Line 29: ...soil δ15N was more effective than plant δ15N in indicating....

Answer: In the revised manuscript, we have indicated that soil $\delta^{15}N$ was more effective than plant $\delta^{15}N$ in indicating the openness of global ecosystem N cycling (P2L31-33).

9. Line 40: cycle -> cycling

Answer: In the revised paper, "cycle" has been replaced by "cycling" (P3L43).

10. Line 48: becomes a useful tool -> is often used

Answer: In the revised paper, "becomes a useful tool" has been replaced by "is often used" (P3L54).

11. Line 68: In addition to soil warming, air warming was also conducted.

Answer: Sorry for this confusion. In the revised paper, "soil warming experiment" has been replaced by "soil and air warming experiments" (P4L75).

12. Line 93: more than 50? Please specify

Answer: Sorry for this confusion. In the revised paper, we have specified 54 (P6L101).

13. Line 99: can be obtained -> were provided

Answer: In the revised paper, "can be obtained" has been replaced by "were provided" (P6L107).

14. Line 109: Did you do resampling with bootstrap? What are the resampling times?

Answer: Yes. In the revised manuscript, we have stated that "Resampling tests were incorporated into our meta-analysis using the bootstrap method (999 random replicates)" (P6L116-118).

15. Line 122: Mean effect sizes: This concept was not specified earlier. I think that it is technique, soil type, warming treatment... . Is it so? Please, clarify

Answer: Sorry for this confusion. For 1 iteration, 1 Hedges' *d* value (effect size) can be obtained. In this case, we can obtain 999 Hedges' *d* values from 999 iterations. The mean effect size is the mean value of Hedges' *d* values.

16. Line 154-159: The authors should add more references to support this statement.

Answer: In the revised paper, we have added more references to support this statement, i.e., Sorensen and Michelsen, 2011; Rousk and Michelsen, 2017; Wang et al., 2018 (P9L171-172).

17. Line 160-164: Is that true? There were two kinds of 15N, three soil acidity-alkalinity types, three vegetation types, and three warming treatments groups. Were there enough number of observations for these groups? For example, for plant δ15N, there were only 9, 3 and 2 observations for acid, neutral and alkali, respectively.

Answer: Although there were only 9, 3 and 2 observations for acid, neutral and alkali, respectively, it will not substantially affect the results of meta-analysis. In previous studies, less than 10 samples can also be used for meta-analysis (Please see below).

(1) Feng, J., Zhu, B., 2019. A global meta-analysis of soil respiration and its components in response to phosphorus addition. Soil Biology and Biochemistry 135, 38–47.

[Figure]

**Fig. 2.** Weighted response ratio of soil respiration ($R_S$), heterotrophic respiration ($R_h$) and autotrophic respiration ($R_a$) in response to phosphorus addition. BGB, AGB, LFP, MBC, SOC, STN, SAN, NON, NHN, STP and SAP represent plant belowground biomass, plant aboveground biomass, litterfall production, microbial biomass carbon, soil organic carbon, soil total nitrogen, soil available nitrogen, soil ammonium nitrogen, soil nitrate nitrogen, soil total phosphorus and soil available phosphorus, respectively. Boreal F, Temp. F and Trop. F represent boreal forest, temperate forest and tropical forest, respectively. Others in soil respiration includes three and two observations from desert and tundra, respectively. Others in heterotrophic respiration included one observation from heath and six observations from unidentified ecosystems according to the descriptions in original case studies. The errors represent 95% confidence intervals (CIs) of weighted response ratio. If 95% CIs did not overlap zero, the P addition effect was considered significant ($p < 0.05$, denoted by *). The numbers in parentheses represent the sample sizes of observations.

(2) Zhang, X.Z., Shen, Z.X., Fu, G., 2015. A meta-analysis of the effects of experimental warming on soil carbon and nitrogen dynamics on the Tibetan Plateau. Applied Soil Ecology 87, 32–38.

[Figure]

**Fig. 2.** Effect sizes of the experimental warming on soil carbon, soil nitrogen, soil microbial biomass carbon (MBC) and nitrogen (MBN), ammonium nitrogen ($NH_4^+$-N) and nitrate nitrogen ($NO_3^-$-N) for alpine forests (a), grasslands (b) and forests + grasslands (c) from a meta-analysis of 25 studies on the Tibetan Plateau. The error bars indicate effect sizes and 95% bootstrap confidence intervals. The warming effect was statistically significant if the 95% CI did not bracket zero. The dashed lines are drawn at effect size = 0. The sample size for each variable is shown next to the bar.

(3) Song, X.Z., Peng, C.H., Zhou, G.M., Jiang, H. & Wang, W.F., 2014. Chinese Grain for Green Program led to highly increased soil organic carbon levels: A meta-analysis. Sci. Rep. 4, 4460; DOI:10.1038/srep04460.

[Figure]

**Figure 3 | Untransformed response ratios pertaining to the effects of the conversion of cropland to forest on soil organic carbon contents at depths of 0–20 cm (a), 20–40 cm (b), and 40–60 cm (c).** Dots with error bars denote the overall mean response ratio and 95% CI. Capital letters to the right of the bars indicate statistically significant differences based on the plantation type or conversion period at the $p < 0.05$ level.

18. Line 167: add a reference here

Answer: In the revised manuscript, we have added a reference here, i.e., Kyveryga et al., 2004 (P9L180).

19. Line 182-184: awkward sentence

Answer: Sorry for this confusion. In the revised paper, we have rewritten this sentence. In the study of Mayor et al. (2015), who found that soil and plant $\delta^{15}N$ were significantly ($p < 0.001$) and negatively correlated with latitude at the global scale. However, the Hedges' $d$ values of soil and plant $\delta^{15}N$ had weak correlations with latitude in this study (Fig. 3) (P10L197-200).

20. Line 185: If you want to use "significantly", you have to give the P value.

Answer: In the revised paper, we have provided the P value. The warming effect on soil $\delta^{15}N$ was significantly ($p < 0.001$) influenced by altitude, MAT and MAP (P10L201).

21. Line 188: Add a reference.

Answer: In the revised manuscript, we have added a reference here, i.e., Craine et al., 2015 (P10L204).

22. Line 211: when investigating -> in order to better investigate

Answer: In the revised paper, "when investigating" has been replaced by "in order to better investigate" (P11L231).

---

## Author Comment (AC3)

**Response to RC2**

1. In the manuscript, 'Soil and plant $\delta^{15}N$ have a different response to experimental warming: A global meta-analysis', the authors assess 20 experimental warming field studies and conclude that soil and plant δ15N had negative and positive responses to warming at the global scale, respectively. Overall, the study is a nice contribution because it looks at both plants and soils. But I think the title oversells and misleads. Also, I realize that this is a short communication, but more detail is needed to support the hypothesis, to rationalize why the specific environmental variables were chosen over others, and to relate this study to other results in the literature.

The word 'significantly' is overused in the abstract, and the presentation of results that are not significant as effects is not appropriate. It would be better to not use the word significantly and to only present the significant results (after defining p-value cut-off in methods). In other words, remove the inference from the title and abstract that plant δ15N had a positive response to warming – this was not significant. I think the finding that soil δ15N is a better indicator than plants of environmental cues is a more appropriate conclusion or story lead. Because really, the pattern of response of plants and soils to environmental drivers tested here was not different, it was just weaker in plants than soil.

Answer: Thank you very much for your efforts on our paper submitted to the "Soil" (Manuscript ID soil-2021-40). We have checked the manuscript and revised it according to the comments carefully.

In the revised manuscript, the title has been changed to "Soil $\delta^{15}N$ is a better indicator of ecosystem nitrogen cycling than plant $\delta^{15}N$: A global meta-analysis". In addition, we also re-conducted the meta-analysis to investigate the influences of soil texture, warming period and increase in temperature on the warming effects on soil and plant $\delta^{15}N$. From Fig. 2gh, the finer the soil texture, the more significant the positive effect of warming on soil and plant $\delta^{15}N$. The possible reason is that the finer the soil texture, the stronger the adsorption of various ions on the soil and the smaller the leaching loss of the soil, resulting in the greater the residual amount of $^{15}N$ in the

soil (Webster et al., 1986). In addition, the longer warming period and the greater increase in temperature resulted in the more negative effect of warming on soil $\delta^{15}N$ (Fig. 2ik). Chang et al. (2017) deduced that N fixation was greater under warming and consequently resulted in a lower soil $\delta^{15}N$ (P10L189-196).

Figure 2:

[Figure]

Finally, we have removed the inference from the title and abstract that plant $\delta^{15}N$ had a positive response to warming – this was not significant. Similar modifications have been made elsewhere in the paper (P2L19-20). We have indicated that a

significant decreasing trend in soil $\delta^{15}N$ and no significant trend in plant $\delta^{15}N$ were found in this study (P8L163-164).

2. l. 44. Please explicitly define 'openness' in the introduction (and if possible, in the abstract). Although some readers will understand, those unfamiliar with the δ15N literature will read this as jargon.

Answer: In the Introduction section of the revised paper, we have explained that openness is a measure of both N inputs and outputs relative to internal cycling and determines both the potential rate of N accumulation in the ecosystem and the potential for N losses following a disturbance (Rastetter et al., 2021) (P3L49-52). However, we did not define "openness" in the abstract due to space limitations.

3. l. 50-51. This should be reversed: The isotopic fractionation effect results in gradual 15N enrichment.

Answer: Sorry for this confusion. In the revised paper, we have indicated that the isotopic fractionation effect results in gradual $^{15}N$ enrichment in the ecosystem (P4L56-57).

4. l. 74-75. The hypothesis is not supported by any rationale in the introduction, please provide some preamble that supports why they should be different.

Answer: Sorry for this confusion. In the revised paper, we hypothesized that soil $\delta^{15}N$ is a better indicator of ecosystem N cycling than plant $\delta^{15}N$ (P5L81-82).

5. l. 85-86. It is not clear why temperature gradient studies are being excluded, as they will also include a treatment and control. Perhaps the authors could clarify - do they mean climate gradients (space-time substitution), or lab incubations?

Answer: Sorry for this confusion. In the revised paper, "temperature gradient" has been replaced by "climate gradients (space-time substitution)" (P5L92-93).

6. l. 114. I am quite surprised that length of warming was not considered, as multiple studies illustrate different responses in plant and soil CNP for short and long-term warming experiments. Also, see Craine et al (2015), which suggests that soil δ15N is directly controlled by soil C and texture, and only indirectly controlled by temperature. Did you consider length of warming, SOC, or soil texture as sub-groups? I see l. 47-67 in the intro provides justification for the subgroups, but I remain unconvinced that the chosen subgroups are more important than the ones not assessed. Perhaps this needs better support from the literature.

Answer: In the revised manuscript, we re-conducted the meta-analysis to investigate the influences of soil texture, warming period and increase in temperature on the warming effects on soil and plant $\delta^{15}N$. From Fig. 2gh, the finer the soil texture, the more significant the positive effect of warming on soil and plant $\delta^{15}N$. The possible reason is that the finer the soil texture, the stronger the adsorption of various ions on the soil and the smaller the leaching loss of the soil, resulting in the greater the residual amount of $^{15}N$ in the soil (Webster et al., 1986). In addition, the longer warming period and the greater increase in temperature resulted in the more negative effect of warming on soil $\delta^{15}N$ (Fig. 2ik). Chang et al. (2017) deduced that N fixation was greater under warming and consequently resulted in a lower soil $\delta^{15}N$ (P10L189-196).

Figure 2:

[Figure]

[Figure]

However, we did not consider the effect of soil organic matter content since only four literatures provided the organic matter content values.

7. l. 176. Actively layer > active layer

Answer: In the revised paper, "actively layer" has been replaced by "active layer" (P9L187).

8. l. 177. This is not true for all air warming treatments.

Answer: Yes. I agree with your comment. In the revised paper, we have indicated that air warming directly impacts aboveground temperatures and has an indirectly effect on soil $\delta^{15}$N (Pardo et al., 2006) (P9L187-189).

9. Table 1 should have a column for soil types – at the very least, organic or mineral soil, but especially pH, since this was a main factor in the analysis.

Answer: In the revised paper, we have provided soil types, soil pH, and organic matter content in Table 1.

10. Many grammatical typos, please correct: l. 40, 42, 43, 49, 68, 83, 85, 171

Answer: Sorry for these errors. In the revised paper, we have corrected them (L43, 45, 46, 55, 75, 90, 184).

---

## Author Response (AR2)

Dear Editor,

Thank you very much for your and reviewers' efforts on our paper submitted to the "Soil" (Manuscript ID soil-2021-40). We have checked the manuscript and revised it according to the comments carefully. The revision has been highlighted in the document by using colored text. We submit here the revised manuscript as well as an itemized response to reviewers' comments.

Sincerely yours,

Dr. Kaihua Liao

**Response to Topical Editor**

1. l. 55. always preferentially loses > is always preferentially lost

Answer: Sorry for this error. In the revised manuscript, "always preferentially loses" has been replaced by "is always preferentially lost" (P3L55).

2. l. 65. remained > remains, showed > shown

Answer: In the revised manuscript, "remained" and "showed" have been changed to "remains" and "shown", respectively (P4L65).

3. l. 68. even that there was no correlation between them > that they were not correlated.

Answer: In the revised manuscript, "even that there was no correlation between them" has been replaced by "that they were not correlated" (P4L68).

4. l. 81-82. The hypothesis has been changed but this does not change my initial comment, that the hypothesis has no support. Please state here why this is your hypothesis. Please make sure that the rationale you provide is the logical conclusion of the information presented in the introduction.

Answer: Thank you for your suggestion. In the revised paper, we have supported the hypothesis. Previous studies (e.g., Liu and Wang, 2009; Wang et al., 2014) have

found that the correlation between soil $\delta^{15}$N and environmental factors was stronger than that for plant, which may be due to the fact that soil samples represented a long-term average for a given location, while plant samples were affected by the microenvironment or the short-term environmental fluctuations. Therefore, we specifically hypothesized that soil $\delta^{15}$N is a better indicator of ecosystem N cycling than plant $\delta^{15}$N (P5L81-85).

5. l. 181. has limit > had limited

Answer: In the revised manuscript, "has limit" has been replaced by "had limited" (P9L186).

6. l. 183. Warming treatment > The type of warming treatment

Answer: In the revised manuscript, "Warming treatment" has been replaced by "The type of warming treatment" (P10L188).

7. l. 188. indirectly > indirect

Answer: In the revised manuscript, "indirectly" has been replaced by "indirect" (P10L193).

8. l. 192. the greater the residual > greater residual

Answer: In the revised manuscript, "the greater the residual" has been replaced by "greater residual" (P10L197).

9. l. 197. remove 'who found that'

Answer: In the revised manuscript, we have removed "who found that" (P10L202).

10. l. 202-207. The direct response of d15N to temperature has already been discussed. Instead it should be discussed here why historical temperature (MAT) influences d15N response to warming.

Answer: In the revised manuscript, we have indicated that the warming effect on soil $\delta^{15}N$ was significantly ($p < 0.001$) influenced by altitude, MAT and MAP. Among these, the strongest correlation was observed for MAT. It is possible that soil $\delta^{15}N$ increased with increasing MAT when the MAT exceeded a certain threshold (e.g., 9.8 $^{o}C$ as proposed by Craine et al. (2015)). In this case, the increase in MAT can enhance the positive effect of experimental warming on soil $\delta^{15}N$. In addition, the MAT can also affect ecosystem N cycle by influencing soil texture. Craine et al. (2015) reported that hot sites had greater clay concentrations than cold sites. As depicted in Fig. 2g, the finer the texture of the soil, the more significant the effect of experimental warming on soil $\delta^{15}N$ (P10L205-213).